# High Light Acclimation Mechanisms Deficient in a PsbS-Knockout Arabidopsis Mutant

**DOI:** 10.3390/ijms23052695

**Published:** 2022-02-28

**Authors:** Young Nam Yang, Thi Thuy Linh Le, Ji-Hye Hwang, Ismayil S. Zulfugarov, Eun-Ha Kim, Hyun Uk Kim, Jong-Seong Jeon, Dong-Hee Lee, Choon-Hwan Lee

**Affiliations:** 1Department of Integrated Biological Science, Department of Molecular Biology, Pusan National University, Busan 46241, Korea; longhoo@naver.com (Y.N.Y.); iszulfugarov@pusan.ac.kr (I.S.Z.); eunhada@korea.kr (E.-H.K.); 2Graduate Department of Life and Pharmaceutical Sciences, Center for Cell Signaling & Drug Discovery Research, Ewha Womans University, Seoul 03760, Korea; lethithuylinh13@gmail.com (T.T.L.L.); hwangjh0012@naver.com (J.-H.H.); lee@ewha.ac.kr (D.-H.L.); 3Institute of Molecular Biology and Biotechnology, Azerbaijan National Academy of Sciences, 11 Izzat Nabiyev Avenue, Baku AZ 1073, Azerbaijan; 4Department of Bioindustry and Bioresource Engineering, Plant Engineering Research Institute, Sejong University, Seoul 05006, Korea; hukim64@sejong.ac.kr; 5Graduate School of Biotechnology, Crop Biotech Institute, Kyung Hee University, Yongin 17104, Korea; jjeon@khu.ac.kr; 6New Mexico Consortium, Los Alamos, NM 87544, USA

**Keywords:** arabidopsis, gene expression, high light acclimation, plastoquinone synthesis, PsbS

## Abstract

The photosystem II PsbS protein of thylakoid membranes is responsible for regulating the energy-dependent, non-photochemical quenching of excess chlorophyll excited states as a short-term mechanism for protection against high light (HL) stress. However, the role of PsbS protein in long-term HL acclimation processes remains poorly understood. Here we investigate the role of PsbS protein during long-term HL acclimation processes in wild-type (WT) and *npq4-1* mutants of *Arabidopsis* which lack the PsbS protein. During long-term HL illumination, photosystem II photochemical efficiency initially dropped, followed by a recovery of electron transport and photochemical quenching (qL) in WT, but not in *npq4-1* mutants. In addition, we observed a reduction in light-harvesting antenna size during HL treatment that ceased after HL treatment in WT, but not in *npq4-1* mutants. When plants were adapted to HL, more reactive oxygen species (ROS) were accumulated in *npq4-1* mutants compared to WT. Gene expression studies indicated that *npq4-1* mutants failed to express genes involved in plastoquinone biosynthesis. These results suggest that the PsbS protein regulates recovery processes such as electron transport and qL during long-term HL acclimation by maintaining plastoquinone biosynthetic gene expression and enhancing ROS homeostasis.

## 1. Introduction

Being sessile organisms, plants have developed sophisticated acclimation mechanisms to cope with unpredictable challenges in their environment. While light is essential for photosynthesis, growth, and reproduction, the exposure of plants to high light (HL) that exceeds their capacity to dissipate the excessive light energy may cause a range of HL stress responses. Light availability can be highly variable and unpredictable in nature. When plants are exposed to excessive HL, the over-excitation of the photosynthetic apparatus triggers short-term protection mechanisms as well as several long-term acclimation processes. As an important short-term HL protection mechanism, non-photochemical quenching (NPQ) dissipates excess excitation energy as heat to reduce oxidative damage [1,2]. Rapidly relaxing qE-type energy-dependent quenching is a fast and major component of NPQ that relies on the development of a trans-thylakoid proton gradient (ΔpH), the accumulation of zeaxanthin involved in the xanthophyll cycle, and the presence of a PsbS subunit of photosystem (PS) II [3,4,5,6,7]. A qE-deficient mutant, *npq4-1* lacking the PsbS protein of PSII was reported in *Arabidopsis* [7]. The role of the PsbS protein in qE generation has now been confirmed in a number of vascular plants including rice PsbS knockout mutants [4,5] and *Populus* PsbS RNAi lines [8]. HL stress ultimately causes the generation of multiple reactive oxygen species (ROS), including hydrogen peroxide (H_2_O_2_), superoxide anion radicals (hereafter superoxide) (O_2_^−^), and singlet oxygen (^1^O_2_) that can cause damage and act as signaling molecules involved in regulating development and pathogen defense responses [9] The long-term protection mechanisms to HL stress or HL acclimation processes include dynamic changes in gene expression involved in hormone biosynthesis, signaling, and photosynthesis, the anthocyanin biosynthesis pathway genes [10], the composition, and the structure of the thylakoid membrane [11,12,13,14], accumulation of antioxidant metabolites and scavenging enzymes [15,16], and reducing LHCII antenna size [17,18,19].

In PsbS knockout rice mutants, a high level of superoxide is accumulated under photo-inhibitory illumination. This may be due to changes in the redox state of the plastoquinone (PQ) pool [5]. Significantly, the redox state of the PQ pool controls many aspects of photosynthesis. Furthermore, higher PQ pool-reduction states were detected in *Arabidopsis* and rice plants lacking the PsbS protein [5,20]. However, the underlying mechanisms of the control of the PQ pool redox state in *psbS* mutants are largely unknown. Plastoquinone-9 (PQ-9) is a photosynthetic electron carrier in chloroplast thylakoid membranes, carrying electrons from PSII to the cytochrome b6/f complex [21]. Oxidized PQ also acts as a ROS scavenger in plant leaves, playing a central photoprotective role [22]. When *Arabidopsis* plants were exposed to long-term HL, the amount of plastoquinone rapidly decreased, followed by a progressive recovery during the acclimation phase [2], indicating a critical role for PQ in HL stress responses.

In this study, we investigated the role of PsbS protein during the HL acclimation processes in *Arabidopsis*. We compared several HL acclimation processes in WT with those in *npq4-1* mutants. To understand the underlying molecular mechanisms involved in the PsbS-dependent HL acclimation process, we carried out gene expression analyses using gene set enrichment analysis (GSEA) systems [23]. GSEA evaluates microarray data at the level of gene sets. This gene set analysis is not useful for discovering new genes, but it is useful in discovering gene sets that have the same or related biological functions [24]. A co-expressed gene set is defined as a cluster of genes that have similar expression patterns under various conditions [23]. From GSEA and qRT-PCR analyses, we observed that the expression of genes involved in PQ-9 biosynthesis was substantially reduced in *npq4-1* mutants during long-term HL illumination resulting in a reduction in the PQ pool size. From these results, we suggest that PQ biosynthesis is one of the acclimation processes induced under HL stress. We propose that the induction of many acclimation processes including PQ biosynthesis is reduced due to the over-accumulation of ROS.

## 2. Results

To monitor the changes in PSII photochemical efficiency (Fv/Fm) of plant leaves under HL illumination, leaves of *Arabidopsis* WT and *npq4-1* mutants grown at low (70 μmol photons m^−2^ s^−1^) photosynthetic photon flux density (PPFD) and were harvested in the middle of the day followed by exposure to HL at 700 μmol photons m^−2^ s^−1^ for up to five days. The temperature at the surface of leaves was set to 15 °C to accelerate the effect of HL stress. During the first 9 h of HL illumination, PSII photoinactivation was rapid, as shown by the rapid drop of Fv/Fm from 0.8 to 0.37 in the WT and 0.8 to 0.17 in the *npq4-1* mutants, respectively (Figure 1A). Afterward, PSII photochemical efficiency slowly increased in the WT plants, as indicated by a progressive increase in Fv/Fm, reaching a value greater than 0.7 after 5 d. However, PSII photochemical efficiency in the *npq4-1* mutants dropped to 0.16 and failed to recover after five days (Figure 1A). During HL illumination (5 d) changes in leaf color were observed indicating anthocyanin accumulation in the WT leaves, but in the *npq4-1* mutants, the changes in leaf color were less obvious, and photobleaching was observed in older leaves (Figure 1B). Figure 1C shows the gradual changes in Fv/Fm during HL illumination for 3 d in whole leaves in the intact WT and the *npq4-1* mutant plants. Similar to detached leaves, the Fv/Fm of the intact leaves of WT dropped substantially until 24 h but recovered after HL treatment, but in the case of the *npq4-1* mutants, leaves that had substantially reduced Fv/Fm ratios did not recover. These results indicated that the WT plants were being damaged before 24-h HL treatment, but after 24-h treatment, the WT plants were entering the acclimation process. However, in the case of the *npq4-1* mutants, damaged plants could not recover and photobleaching occurred.

To address the role of NPQ in the HL acclimation process, we measured the NPQ light response curves in WT and *npq4-1* plants during HL treatments. After HL illumination for 9 h, the NPQ values in WT became higher than in non-treated control leaves, but the NPQ values dropped significantly following HL illumination for 24 h and further decreased after 48 h HL treatment (Figure 2A). As shown in Figure 2B, the degree of NPQ development in *npq4-1* plants exposed to variable illumination intensities was largely impaired during HL illumination presumably due to the loss of the PsbS protein [3]. These results show that the PsbS protein plays an important role in HL stress amelioration during short-term HL treatments, but that PsbS is not an essential factor during long-term HL treatments.

The major component of NPQ, energy-dependent quenching (qE) includes the accumulation of the xanthophyll cycle pigment zeaxanthin associated with the development of the pH gradient across the thylakoid membrane and the PsbS protein [7,25]. We compared the rise in kinetics of de-epoxidation state of the xanthophyll cycle pigments in WT and *npq4-1* mutant plants. As shown in Figure 3A, there was almost no noticeable difference in the de-epoxidation rise kinetics between WT and *npq4-1* mutants. This is not surprising since xanthophyll cycle activities in *npq4-1* mutants are reported to be fully functional [7,26]. Additionally, there was no change in the size of the xanthophyll pool during the HL treatment for three days and no noticeable difference between WT and *npq4-1* (Figure 3B).

Because qE develops during photosynthetic electron transport, light intensity dependent changes in electron transport rates (ETR) were compared between WT and *npq4-1* mutants (Figure 4). In non-HL-treated plants, the light intensity dependent ETR curve in WT plants was very similar to *npq4-1* mutants. After HL illumination for 9 h, the ETR curves in both WT and *npq4-1* mutants were severely reduced. However, reduced ETRs were partially recovered during acclimation for 24 h and further after 48 h treatment, but no ETR recovery was observed in *npq4-1* mutants.

A chlorophyll fluorescence parameter, qL, was measured to access the fraction of open PSII centers, newly derived from the lake model [27]. As shown in Figure 5, there was no difference in qL values between WT and *npq4-1* mutants before HL illumination. However, the qL values decreased rapidly during the HL treatment period for 9 h in WT plants. Interestingly, qL decreased even further in *npq4-1* mutants than in WT plants. As observed here, under the photo-inhibitory condition, the fraction of open PSII centers, qL, decreases which reflects the rapid changes in the redox level of the PQ pool. After HL illumination for 24 h, WT plants entered into an acclimation phase associated with the gradual recovery of qL values, but qL values in *npq4-1* plants did not recover at all.

Exposure to excessive light in both WT and *npq4-1* mutants resulted in the production of ROS [5,20]. The accumulation of superoxide was visualized by histochemical staining of the whole plant of *Arabidopsis* with nitroblue tetrazolium (NBT) (Figure 6A—top panels). In low-light-grown control plants, no noticeable difference was observed between WT and *npq4-1* mutants. However, in the case of HL-treated plants, *npq4-1* plants stained darker than WT plants, suggesting that more superoxide was accumulated in *npq4-1* mutants compared to WT. Because superoxide is rapidly dismutated to more stable hydrogen peroxide by SOD [28], hydrogen peroxide production was measured by histochemical staining with 3, 3′-diaminobenzidine (DAB) (Figure 6A—bottom panels). DAB test results were very similar to those from the superoxide accumulation test, suggesting that more hydrogen peroxide was accumulated in *npq4-1* mutants compared to WT following HL stress. Singlet oxygen is also a photosynthesis byproduct that is mainly formed from PSII under HL conditions, and which may damage the photosynthetic apparatus [29]. Therefore, singlet oxygen production was also measured in WT and *npq4-1* leaves using the singlet oxygen sensor green (SOSG) [30,31]. In HL-untreated control leaves, the kinetics for increased SOSG fluorescence emission induced by photo-inhibitory illumination in *npq4-1* plants was similar to the kinetics in WT. However, in the case of HL-treated leaves, singlet oxygen production increased in both WT and *npq4-1* leaves but was greater in *npq4-1* leaves than in WT leaves (Figure 6B).

Changes in the chlorophyll a/b ratio as a result of acclimation to a fluctuating light environment have been used as a measure of the changes in the relative LHCII contents in chloroplasts of vascular plants and microalgae [32,33,34]. When *Arabidopsis* WT plants were transferred to HL conditions (700 μmol photons m^−2^ s^−1^), the chlorophyll a/b ratio increased from 2.8 to 3.4 (Figure 7A), which is in accordance with previous results [33,35,36,37]. However, under the same conditions, *npq4-1* mutants showed a more rapid increase in the chlorophyll a/b ratio from 2.8 to 3.2 at the beginning of HL treatment, but during the acclimation phase, chlorophyll a/b ratios dropped and the total chlorophyll amount decreased faster than WT (Figure 7B). The total chlorophyll content (per g fresh weight) in both plants decreased during the HL treatment for 3 d, however, the loss of Chl in *npq4-1* mutants was faster than in WT plants (Figure 7B).

To capture any differences in gene expression during HL acclimation, we analyzed mRNA pool sizes for various genes in WT and *npq4-1* mutants after HL treatments for either 3 h or 24 h by microarray analysis. Microarray results were analyzed by the Gene Set Enrichment Analysis (GSEA) program, and the results were visualized on a co-expression terrain map with 27 co-expressed gene sets (Figure 8A, here, each of 27 gene sets was named according to its position in the panel shown in the left-top corner). Since gene sets that were relatively closely related in their expression patterns converged into one ridge, the numbers of gene sets on the terrain map did not always correlate to the numbers of enriched gene sets in each analysis. We used pairwise comparisons (e.g., 0 h of WT-HL versus 3, 24 h of WT-HL or 0, 3, 24 h of *npq4-1*-HL) to identify differentially expressed genes at each time point in WT and the *npq4-1* mutants (as shown in five panels in Figure 8A). In each panel, gene sets positively enriched in each treatment are visualized by red square dots, and conversely gene sets negatively enriched are represented by blue square dots. For the 27 co-expressed gene sets, the color code for each gene set is tabulated in Figure 8B. The short-term HL treatment for 3 h lead to the identification of six co-expressed gene sets that were positively enriched in WT. Only three of the gene sets remained positively enriched after HL treatment for 24 h. In *npq4-1* mutants treated with HL, the enriched gene sets were different from those in WT except for the F16 gene set, which was also positively enriched in HL-treated mutants for 24 h. Two gene sets, G14 and L09, were interesting for us in that these gene sets were positively enriched in both WT plants treated with HL for 3 h and 24 h, but not in HL-treated *npq4-1* mutants. The lists of genes are tabulated in Appendix A for G14 and in Appendix A for L09.

The genes encoding *Solanesyl Diphosphate Synthase 2* (*SPS2*) and a tocopherol cyclase (*Vitamin Deficient 1*; *VTE1*), involved in PQ and tocopherol biosynthesis, were notable in the G14 gene set. Both enzymes are involved in PQ biosynthesis and knock-out mutants have been shown to be sensitive to high light [2,38]. Recently, SPS-over-expression plants have been shown to accumulate higher amounts of PQ during HL acclimation and show tolerance to high light [2]. To gain a better understanding of how gene expression is regulated by HL and to confirm microarray data, we analyzed the expression of PQ biosynthesis genes in WT and *npq4-1* mutants during HL stress by qPCR. As previously reported [2], known HL-induced PQ biosynthesis genes (e.g., *SPS1*, *SPS2*, *VTE1* and *VTE3*) were highly induced during HL stress in WT, but the induction of PQ biosynthesis genes was very slow in *npq4-1* compared with that in WT (Figure 9). These expression patterns confirm the results obtained from the microarray analysis.

## 3. Discussion

In order for plants to survive in constantly changing and/or stressful environments they must be able to alter their metabolism to reduce and manage potentially stress-related damage. Light is one of the fastest-changing abiotic stress factors. When light levels exceed the capacity of photosynthesis, the excess absorbed energy can lead to the inactivation of the reaction centers (RCs) largely due to the formation of reactive oxygen species (ROS) which leads to a reduction in electron transport efficiency and photosynthesis as a whole—the phenomenon known as photo-oxidation or photo-inhibition [39,40].

Both photochemical mechanisms and non-photochemical protection mechanisms have been shown to reduce damage to PSI and PSII. The former mechanisms lead to enhanced electron sink strengths to prevent the over-reduction of electron carriers in the photosynthetic electron transfer chain, including the activation of key Calvin–Benson cycle enzymes, which can be easily monitored by measuring ETR. The latter mechanisms involve the dissipation of excess energy absorbed by the photosystems as heat, which can be easily monitored by measuring NPQ. Under photo-inhibitory conditions, some parts of the photosynthetic apparatus cannot be adequately protected by short-term protection mechanisms and are continuously damaged. The damage, however, can be repaired rapidly to continue photosynthetic activities. In addition to short term HL stress protection systems, longer-term repair systems are also engaged to maintain photosynthetic efficiency. In this study, we show that PsbS-deficient *Arabidopsis npq4-1* mutants could not adapt to HL stress as well as WT plants as shown by their reduced ability to recover PSII photochemical efficiency or Fv/Fm relative to WT plants (Figure 1). To understand the molecular basis for this reduced repair capacity in the PsbS-deficient mutant, we compared known, as well as less well known, characterized protection or acclimation mechanisms in WT and *npq4-1* mutants.

Acclimation to high light stress also involves physiological and molecular alterations of the photosynthetic membrane, such as an increased chlorophyll a/b ratio, reduced light-harvesting antenna size, increased anthocyanin content, higher content of the cytochrome b_6_f complex and fewer antenna complexes LHCII and CP24 [19,33,41,42,43,44,45]. In this study, as a sign of high light stress acclimation, we observed that chlorophyll degradation stopped after HL illumination for 24 h in WT, but in *npq4-1* mutants chlorophyll degradation continued to decrease for 5 d (Figure 7B). Together with chlorophyll degradation, the LHCII levels in both plants were reduced as shown in Figure 7A [45,46]. In addition, the reduction in PSII antenna size slowed down after 24 h of HL illumination. This result suggests that the reduction of antenna size is an important HL acclimation mechanism, but the PsbS deficient mutants failed to acclimate by reducing antenna size.

Long-term HL acclimation processes also include the accumulation of antioxidant metabolites including chlorophyll intermediates and anthocyanin and the activation of ROS scavenging enzymes [16,47]. As shown in Figure 6, when plants were grown in HL for 2 h, the amounts of three different ROS species were lower in WT compared with the *npq4-1* mutants, suggesting that WT can acclimate well to reduce ROS production, but not *npq4-1* mutants.

Processes related to non-photochemical ways of protection did not show any sign of recovery after HL illumination for 48 h in WT leaves (Figure 2A). In addition, there were no significant differences between WT and *npq4-1* mutants in the changes both in the pool size and in the de-epoxidation state of the xanthophyll cycle pigments (Figure 3). The drop of NPQ shown after HL illumination for 24 h in WT leaves may be due to the photo-oxidation of PSII, and NPQ could be recovered slowly after the repair of damaged photosystems.

Photochemical protection mechanisms were also activated after 24 h of HL illumination in WT plants as shown by the changes in ETR (Figure 4A), but not in *npq4-1* mutants. When plants were exposed to high light intensity, the size of the electron sink limits photosynthesis. When the electron sinks become saturated, there is an over accumulation of electrons in the photosynthetic electron transport system as observed by the reduction in PQ pool size [27]. When the value of qL of chlorophyll fluorescence that reflects the reduction of PQ was measured, qL also significantly decreased during HL illumination in both WT and *npq4-1*, but the recovery in qL was observed only in WT plants, and not in *npq4-1* mutants (Figure 5). The long-term HL acclimation processes includes the accumulation of plastoquinone [2], which can quench ^1^O_2_ and inhibit oxidation of lipid membranes [48,49,50]. Under HL stress, a dramatic reduction of total PQ-9 content is reported [2], as we have noticed a significant drop in ETR and qL after HL illumination (Figure 4 and Figure 5). Several papers have also reported the accumulation of PQ-9 as a long-term HL acclimation process [38,49,51,52].

The evidence of transcriptional regulation of PQ accumulation comes from the results of GSEA of cDNA microarray analyses. Interestingly, two gene sets including G14 and L09 are positively enriched in 3 h and 24 h HL-treated WT, not in *npq4-1* mutants. The genes *SPS2* and *VTE1* in the G14 gene set are involved in PQ and tocopherol biosynthesis. These genes related to PQ biosynthesis enzymes were upregulated during HL treatment in WT, whereas these genes were not upregulated in *npq4-1* mutants (Figure 9A). These results suggest the possibility that the *npq4-1* mutation affects the synthesis of PQ at the transcriptional level during HL acclimation. The function of PQ goes beyond its traditional roles as photosynthetic electron carriers between PSII and PSI and redox signals regulating cellular activities [45,46]. Longer-term acclimation to HL was associated with a marked rise in the PQ-9 levels which is followed by the recovery of PSII activity.

In this study, our results also suggest that long-term HL acclimation processes include the recovery of PSII activity with an associated increase in ETR and qL. The increase in ETR and qL is accompanied by the rise in the expression of genes involved in PQ biosynthesis. In addition, we could observe other processes known to be involved during HL acclimation including the regulation in ROS homeostasis and a reduction in light harvesting antenna size. However, *npq4-1* mutants failed to exhibit most of these acclimation processes, especially the expression of genes involved in PQ. Signals that initiate HL acclimation include ROS accumulation, which triggers the expression of PQ biosynthesis genes, such as *SPS* and *VTE1* [2,22]. Because the redox state of the PQ pool affects almost all aspects of the photosynthesis including gene expression, the failure of transcription induction of PQ biosynthesis genes in *npq4-1* mutants may interfere with the light stress acclimation processes including antenna size reduction. Upstream signals for the start of the HL acclimation processes may not be properly triggered in the PsbS-deficient *npq4-1* mutants.

## 4. Materials and Methods

### 4.1. Plant Materials, Growth and HL Treatment Conditions

Arabidopsis thaliana WT (ecotype Col 0) and *npq4-1* mutant plants were grown in soil in a growth chamber with a 16 h photoperiod at a photosynthetic photon flux density (PPFD) of 70 μmol photons m^−2^ s^−1^ with a day/night temperature cycle of 23/18 °C. For HL acclimation, 4 to 5-week-old plants were treated with HL at a PPFD of 700 μmol photons m^−2^ s^−1^ at 19 °C on the surface of adaxial side of the leaf.

### 4.2. Chlorophyll Fluorescence

Chlorophyll fluorescence was measured using a PAM-2000 portable chlorophyll fluorometer (PAM2000, Walz, Effeltrich, Germany). The minimum fluorescence at open PSII centers in the dark-adapted state (Fo) was excited by a weak measuring light (wavelength 650 nm) at a PPFD of 0.05–0.1 μmol photons m^−2^ s^−1^. A saturating pulse of white light with PPFD of 3000 μmol photons m^−2^ s^−1^ for 800 msec was applied to determine the maximum fluorescence at closed PSII centers in the dark-adapted state (Fm) and the maximum fluorescence at closed PSII centers in the actinic light (AL)-adapted state (Fm′). The photochemical efficiency of PSII (Fv/Fm) was calculated using the equation, Fv/Fm = (Fm − Fo)/Fm. ETR was computed as previously described as: ETR = (ΔF/Fm′) × PAR × 0.5 × 0.84 (ΔF = (Fm′ − Ft)) assuming equal distribution of excitation between PSI and PSII [53]. The parameter qL was determined as described previously [27,54].

### 4.3. Thylakoid Membrane Isolation

Thylakoid membranes were prepared according to the method described in [55] with some modifications. Detached leaves were ground with a glass homogenizer in ice-cold grinding buffer (50 mM HEPES, pH 7.6, 0.3 M sorbitol, 10 mM NaCl, 5 mM MgCl_2_). The homogenate was filtered through two layers of Miracloth (Merck, Darmstadt, Germany) and centrifuged at 20,000× *g* in a microcentrifuge (Micro17R, Hanil, Daejeon, Korea) for 7 min. The pellet was then washed twice and resuspended in resuspending buffer (50 mM HEPES, pH 7.6, 0.1 M sorbitol, 10 mM NaCl, 5 mM MgCl_2_). The suspension was kept on ice in the dark until use after chlorophyll concentrations were measured according to [56].

### 4.4. Measurement of Photosynthetic Pigments by High Performance Liquid Chromatography

Photosynthetic pigments were analyzed according to the method of Gilmore and Yamamoto [57]. Briefly, leaf segments from one month old plants grown in a greenhouse under a 500 µmol photons m^−2^ s^−1^ light intensity were collected. The fresh weight of each sample was recorded before dark-adaptation for 3 h. Dark-adapted leaf discs were used as control samples, whereas HL illumination of pre-dark-adapted leaf discs was for 3 h at 2000 µmol photons m^−2^ s^−1^. Dark-adapted and HL illuminated samples were immediately frozen in liquid nitrogen and stored at −80 °C until use for pigment extraction. To extract pigments, frozen leaf segments were ground with a Mixer-Mill (Qiagen, Hilden, Germany). The resulting leaf powder was gently agitated in ice-cold acetone for 1 h. To minimize pigment degradation, extraction was performed in darkness at 4 °C. Cell debris was removed by centrifugation twice at 19,000× *g* for 10 min at 4 °C. Extracts were filtered through a 0.2-mm syringe filter and pigments were separated using an HPLC system (Agilent, Santa Clara, CA, USA) equipped with Spherisorb ODS-1 columns (Altech, Nicholasville, KY, USA). The pigment concentration was estimated by using the conversion factors for peak areas (in nmol) that were previously calculated for this solvent mixture [57].

### 4.5. Histochemical Staining of Superoxide and Hydrogen Peroxide

Histochemical staining for ROS production was conducted as previously described [58,59,60], with some modifications. For superoxide determinations, leaf samples were immersed in 6 mM NBT solution containing 50 mM sodium phosphate (pH 7.5) for 12 h in the dark. To detect hydrogen peroxide, detached leaves were immersed in 5 mM DAB solution containing 10 mM MES (pH 3.8) for 12 h under darkness. After light treatment for 2 h, leaves were then decolorized by immersing them in 70% ethanol overnight to thoroughly remove the chlorophyll. The cleared leaves were preserved in 50% ethanol.

### 4.6. Fluorometric Detection of Singlet Oxygen

For the detection of singlet oxygen, leaves were vacuum infiltrated with 200 μM SOSG (Thermo Fisher Scientific, Waltham, MA, USA) in 50 mM phosphate buffer, pH 7.5. While illuminated with light, the SOSG fluorescence emission was measured at 530 nm with excitation at 480 nm using F4500 fluorescence spectrophotometer (Hitachi, Toyoto, Japan).

### 4.7. RNA Isolation and Purification

For microarray experiments, WT and *npq4-1* mutants were grown under continuous light for 3 weeks at a light intensity of 70 μmol photons m^−2^ s^−1^ at 23 °C to avoid the effect of circadian rhythm on gene expression. Total RNA was isolated using RNeasy Plant Mini Kit (Qiagen, Hilden, Germany), following the manufacturer’s instructions. Briefly, the leaf samples of high-light-treated plants were homogenized in the presence of liquid nitrogen and were lysed in a buffer containing guanidine isothiocyanate. The lysed samples were placed in the RNeasy column and washed with an ethanol-containing buffer. Total RNA was eluted with RNase-free water. For ethanol precipitation, 1 mL of 95% ethanol and one-tenth volume of 3 M NaOAc, pH 5.2 were added and the total RNA sample was held at −80 °C for 20 min. After centrifuging at 12,000× *g* for 15 min at 4 °C, the entire RNA pellet was washed with 1 mL of 70% ethanol and centrifuged at 12,000× *g* for 5 min at 4 °C. The pellet was dissolved in RNase-free water. The concentration and purity of this isolated total RNA were determined by measuring absorbances at 260 nm and 280 nm.

### 4.8. Microarray Experiment, Image Acquisition, Data Acquisition and Normalization

Total RNA (5 g) of leaf samples of *A. thaliana* (Col-0) after treatment of HL at 700 µmol photons m^−2^ s^−1^ for 0, 3 and 24 h. were reverse-transcribed using reverse transcription (RT) primer tagged with either Cy3-3DNA or Cy5-3DNA capture sequence of Array 900 MPX Expression Array Detection Kits (Genisphere, Hatfield, PA, USA). The synthesized cDNA including capture sequence were fluorescently labeled by Cy3-3DNA or Cy5-3DNA based on the sequence complementary to the 3DNA capture reagent, which contained an average of 900 fluorescent dyes. The labeled cDNA was hybridized on an Operon Arabidopsis Version 3.0 microarray which consists of synthetic 70-mer oligonucleotides on aminosilane-coated slides by the David Galbraith Lab, University of Arizona. The hybridization and washing procedures were performed according to the Genisphere technical protocol. After washing, the slides were immediately scanned using an ArrayWoRx (Applied Precision, Issaquah, WA, USA). To maximize the camera’s dynamic range without saturation and to normalize the two channels for signal intensity, the exposure setting was adjusted so that the intensity level of the brightest spot on a slide was 80 to 90%. Experiments were performed with three replicates, which generally used the same material except for the dye labels, which were reversed between duplicate samples for each microarray. Intensity values were quantified from the pairs of TIFF image file from each channel using version 5.6 ImaGene software (BioDiscovery, El Segundo, CA, USA). Analyses were using the version 4.1 GeneSight software package (BioDiscovery, El Segundo, CA, USA). For each slide, the local background was subtracted from the signal intensity and the minimum intensity was raised to 20 by using the ‘floor’ function. The mean intensity for each element was normalized by the locally weighted scatterplot smoothing (LOWESS) method in GeneSight software.

### 4.9. GSEA Analysis Using Chloroplast-Specific Coexpression Gene Set Database

The GSEA was performed using version 2.0.7 of the GSEA-P software downloaded from GSEA website (http://www.broadinstitute.org/gsea/, accessed on 12 December 2015) [23]. To calculate the significance of the enrichment score (ES), class labels were randomly permuted, and ES were recalculated 1000 times. In this study, the cutoff for significance of ES was defined as the score according to *p* value of 0.05 and FDR value of 0.25. A statistically significant value for the gene sets represented by less than 10 genes was defined as *p* value of 0.1 since a small population size has a negative influence on statistical significance. GSEA evaluated a query microarray data set by using chloroplast-specific co-expression gene set (CC gene set) database.

### 4.10. Gene Expression Analysis

Total RNAs were extracted using TRIzol reagent and reverse-transcribed into cDNAs using the PrimeScript RT reagent kit (TaKaRa, Tokyo, Japan). RT-qPCR was performed using KAPA SYBR FAST qPCR master mix (Kapa Biosystems, Wilmington, MA, USA) with gene-specific primers on a LightCycler 480 system (Roche, Bâle, Switzerland) according to the manufacturer’s protocol. For transcript normalization, Actin1 was used as a reference gene. Data were analyzed using LC480Conversion and LinRegPCR software (Heart Failure Research Center).

### 4.11. Statistical Analysis

Statistical analyses were performed using GraphPad Prism (v. 8.0). Nonlinear regression analysis was performed by least square methods. Significant differences between experimental groups were analyzed by one-way ANOVA with Fisher’s LSD test or Student’s *t*-test, respectively. Detailed information about statistical analysis is described in the Figure legends. Statistical significance was set at *p* < 0.05. All experiments were repeated three to five times with similar results.

## 5. Conclusions

Plant leaves dissipate excessive light as heat by NPQ. A major part of NPQ, qE, has been studied extensively as a short-term protection mechanism against photoinhibition. However, long-term high light acclimation processes are poorly understood. In this study, it was demonstrated that a qE deficient mutant, *npq4-1* lacking PsbS protein, failed to acclimate well to long-term HL treatment. In WT plants, photosynthesis-related parameters recovered well during long-term HL, while this was not the case for *npq4-1* mutants. LHCII antenna size reduction and ROS homeostasis, which occurred during the HL acclimation process in WT plants, were impaired in *npq4-1* mutants. This was associated with a reduction in the expression of PQ biosynthesis genes in *npq4*-1 mutants during HL treatment. Taken together, it was found that NPQ, which was considered important for short term-HL acclimation, is also an important mechanism for long-term HL acclimation processes, as is the expression of PQ biosynthesis genes.

## Figures and Tables

**Figure 1 ijms-23-02695-f001:**
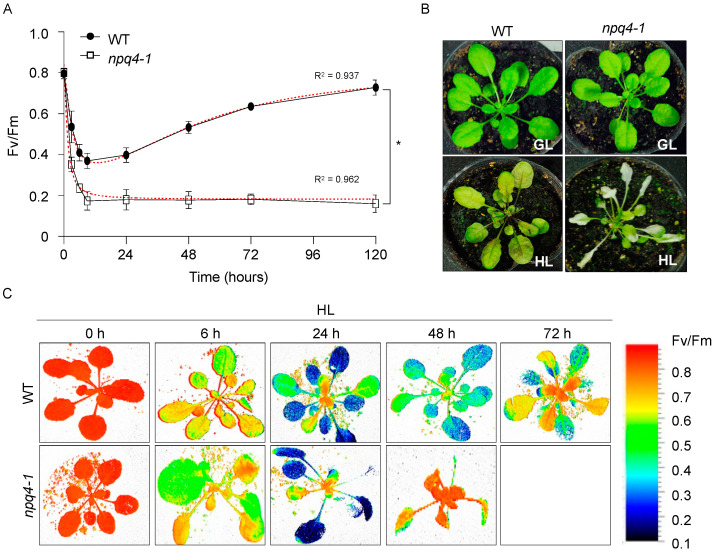
Changes in Fv/Fm and visible phenotypes of WT and *npq4-1* mutants during high light illumination. (**A**) Changes in the photochemical efficiency of PSII (Fv/Fm) during high light (HL) treatment for up to 5 days. The means and standard errors (±SE) from three to five replicates are shown. Regression line plotted as red dashed line estimated via nonlinear least square regression. R^2^ denotes the regression coefficient. Significance differences of fitted curves are indicated by asterisks (* *p* < 0.05) compared to the WT as determined by unpaired Student’s *t* test; (**B**) The visible phenotype of the WT and *npq4-1* mutants grown under growth light (GL) at 70 μmol photons m^−2^ s^−1^ for 4 weeks after transfer to HL at 700 μmol photons m^−2^ s^−1^ for 3 d; (**C**) Chlorophyll fluorescence images for Fv/Fm. The bar on the right side is the color scale for Fv/Fm.

**Figure 2 ijms-23-02695-f002:**
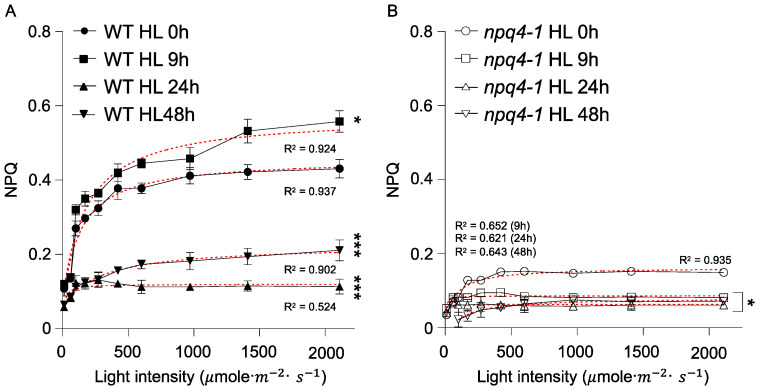
Light response curves for non-photochemical quenching in wild-type and *npq4-1* mutants during high light illumination. Leaves of 4-weeks-old seedlings grown in a growth chamber were treated with high light (HL) at 700 μmol photons m^−2^ s^−1^ for 0, 9, 24, or 48 h and dark-adapted for 10 min before the measurement of non-photochemical quenching (NPQ). Each photosynthetically active irradiance of 0, 70, 110, 180, 250, 400, 550, 900, 1400, or 2200 μmol photons m^−2^ s^−1^ was applied for 10 min for the measurement in (**A**) WT; and (**B**) *npq4-1* mutant leaves. NPQ was calculated as described in Section 4. Each point represents the mean of at least four experiments (SD indicated by the bar). Regression line plotted as red dashed line estimated via nonlinear least square regression. R^2^ denotes the regression coefficient. Significance differences of fitted curves are indicated by asterisks (* *p* < 0.05; *** *p* < 0.001) compared to the 0 h as determined by one-way ANOVA with Fisher’s LSD test.

**Figure 3 ijms-23-02695-f003:**
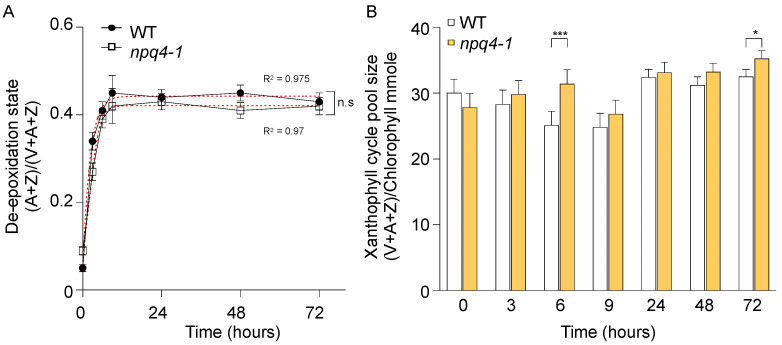
Changes in xanthophyll cycle pigment de-epoxidation state and pigment pool size in wild-type and *npq4-1* mutants during high light illumination. (**A**) Xanthophyll cycle pigment de-epoxidation state is calculated as (A + Z)/(V + A + Z), where V is violaxanthin, A is antheraxanthin, and Z is zeaxanthin. Data are the means ±SE (*n* = 3). Regression line plotted as red dashed line estimated via nonlinear least square regression. R^2^ denotes the regression coefficient. Significance differences of fitted curves are indicated by asterisks (* *p* < 0.05; *** *p* < 0.001) compared to the WT as determined by unpaired Student’s *t* test; (**B**) Total xanthophyll cycle pigments in WT and *npq4-1* mutants during HL treatments. Pool size was calculated by total chlorophyll a + b. Pigment was extracted by 80% acetone and analyzed by HPLC using absorbance at 460 nm. Significance differences are indicated by asterisks (* *p* < 0.05; *** *p* < 0.001) compared to the WT as determined by one-way ANOVA with Fisher’s LSD test.

**Figure 4 ijms-23-02695-f004:**
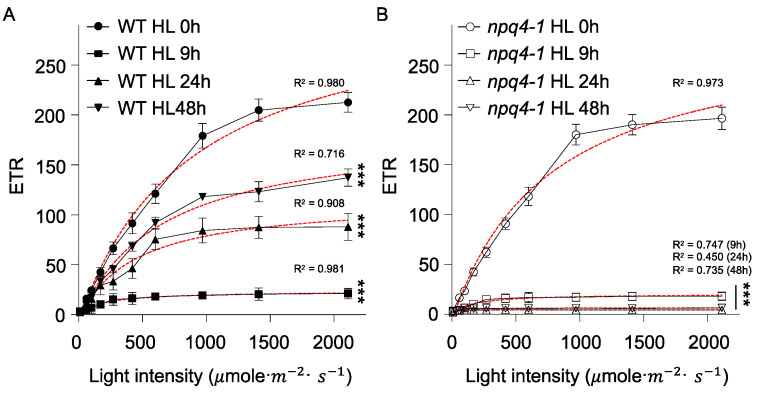
Light response curves for electron transport rate in wild-type and *npq4-1* mutants during high light illumination. Leaves of 4-weeks-old seedlings grown in a growth chamber were treated with high light (HL) at 700 μmol photons m^−2^ s^−1^ for 0, 9, 24, or 48 h and dark-adapted for 10 min before the measurement of electron transport rate. Each photosynthetically active irradiance of 0, 70, 110, 180, 250, 400, 550, 900, 1400, or 2200 μmol photons m^−2^ s^−1^ was applied for 10 min for the measurement in (**A**) WT; and (**B**) *npq4-1* mutant leaves. The electron transport rate was calculated as described in Section 4. Each point represents the mean of at least four experiments (SD indicated by the bar). Regression line plotted as red dashed line estimated via nonlinear least square regression. R^2^ denotes the regression coefficient. Significance differences of fitted curves are indicated by asterisks (*** *p* < 0.001) compared to the 0 h as determined by one-way ANOVA with Fisher’s LSD test.

**Figure 5 ijms-23-02695-f005:**
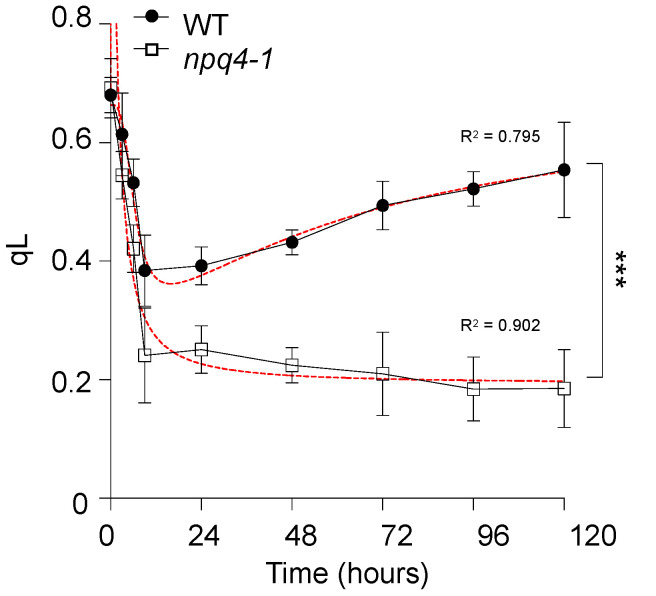
Changes in the fraction of open PSII centers, qL, in wild-type and *npq4-1* mutants during high light illumination. Leaves of 4-weeks-old seedlings grown in a growth chamber were treated with high light (HL) at 700 μmol photons m^−2^ s^−1^ and dark-adapted for 10 min before the measurement of qL. The parameter qL was calculated as described in Section 4. Each point represents the mean of at least four experiments (SD indicated by the bar). Regression line plotted as red dashed line estimated via nonlinear least square regression. R^2^ denotes the regression coefficient. Significance differences of fitted curves are indicated by asterisks (*** *p* < 0.001) compared to the WT as determined by unpaired Student’s *t* test.

**Figure 6 ijms-23-02695-f006:**
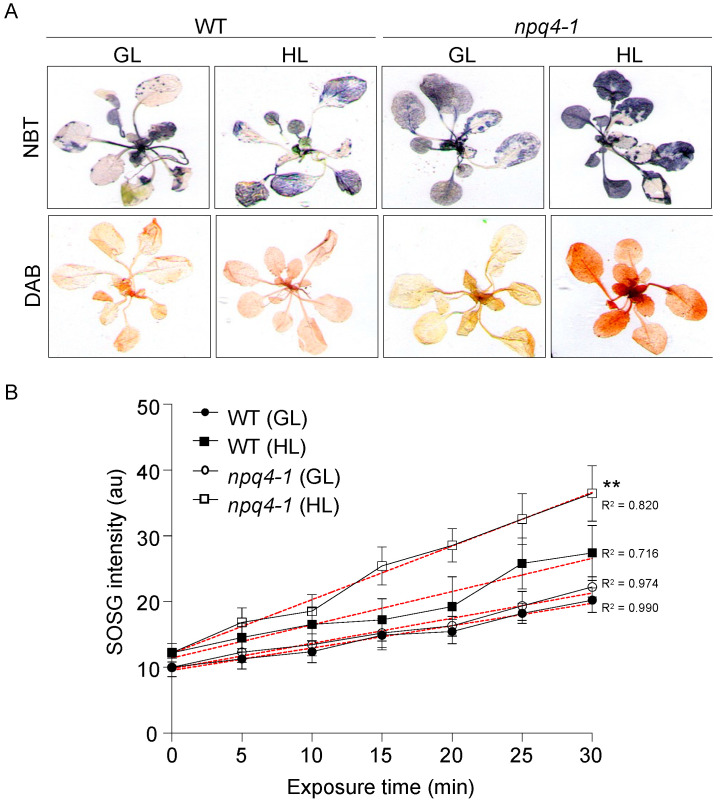
ROS production in wild-type and *npq4-1* mutants under growth light and high light illumination. (**A**) Production of superoxide anion radicals and hydrogen peroxide was detected by histochemical staining with NBT (top panel) and DAB (bottom panel) in wild-type (WT) and *npq4-1* leaves, respectively. Leaves infiltrated with 6 mM NBT or 5 mM DAB solution by immersion for 12 h in darkness were treated with growth light (GL) at 70 μmol photons m^−2^ s^−1^ or high light (HL) at 700 μmol photons m^−2^ s^−1^ for 2 h. The leaves were decolorized by immersing them in 70% ethanol. Experiments were repeated 4–6 times and representative images are shown; (**B**) Detection of singlet oxygen in leaves, as monitored by the increase in SOSG fluorescence emission at 530 nm. Leaf segments were vacuum infiltrated with 200 μM SOSG solution before being illuminated for 30 min with GL at 70 μmol photons m^−2^ s^−1^ or with HL at 700 μmol photons m^−2^ s^−1^. Regression line plotted as red dashed line estimated via nonlinear least square regression. R^2^ denotes the regression coefficient. Significance differences of fitted curves are indicated by asterisks (** *p* < 0.01) compared to the 0 h as determined by one-way ANOVA with Fisher’s LSD test.

**Figure 7 ijms-23-02695-f007:**
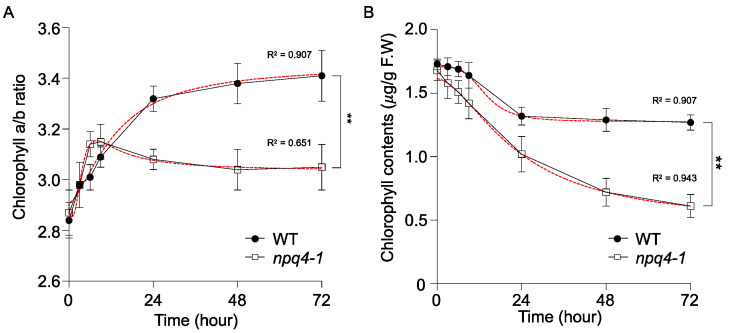
Change in chlorophyll a/b ratio and chlorophyll contents in wild-type and *npq4-1* mutants during high light illumination. (**A**) Chlorophyll a/b ratio reflects LHCII proportion of photosystem during the HL treatment; (**B**) Chlorophyll contents during HL treatment were expressed as μg (g FW)^−1^. Fresh weight was determined after HL treatment. Regression line plotted as red dashed line estimated via nonlinear least square regression. R^2^ denotes the regression coefficient. Significance differences of fitted curves are indicated by asterisks (** *p* < 0.01) compared to the WT as determined by unpaired Student’s *t* test.

**Figure 8 ijms-23-02695-f008:**
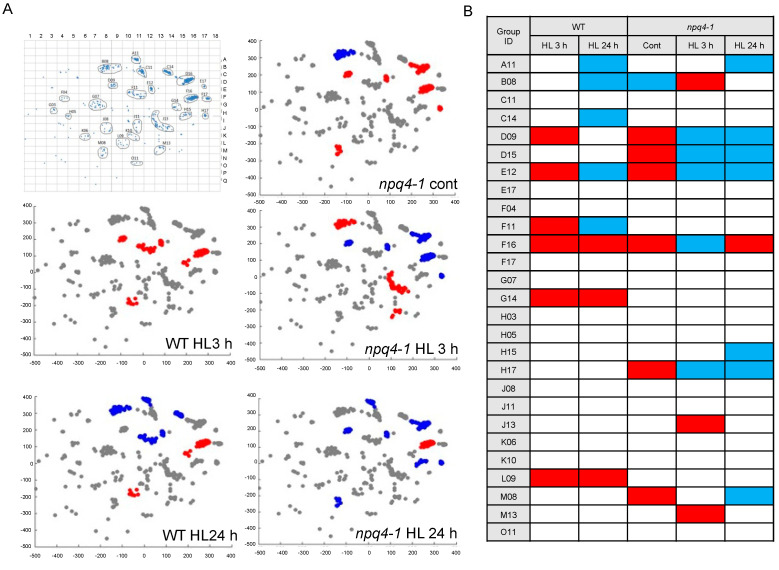
The co-expressed gene sets enriched in wild-type and *npq4-1* mutants during high light illumination. (**A**) The position on the co-expression terrain map of differentially co-expressed gene sets is represented. Gene sets positively enriched for each treatment are represented in red color and conversely, gene sets negatively enriched are represented in blue color. The x, y coordinates of the gene sets on the co-expression terrain map correspond to the names of the gene sets; (**B**) Gene sets were visualized by heat map. WT, wild-type; MT, mutant; HL, high light.

**Figure 9 ijms-23-02695-f009:**
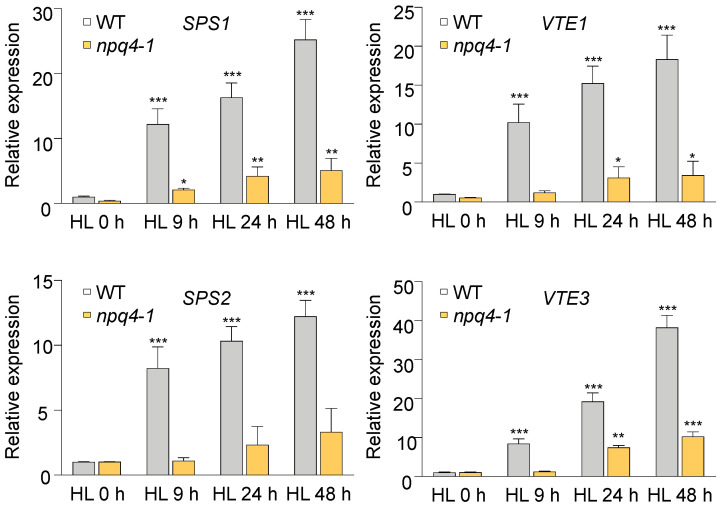
RT-qPCR analysis of PQ synthesis gene expression in WT and *npq4-1* plants in response to HL. Four-week-old Col-0 and *npq4-1* plants were treated with PPFD of 700 μmol photons m^−2^ s^−1^ at 19 °C on the surface of adaxial side of the leaf. Values represent means ± SD (*n* = 3 biological replicates). Significance differences are indicated by asterisks (* *p* < 0.05, ** *p* < 0.01, *** *p* < 0.001) compared to the HL 0 h WT as determined by one-way ANOVA with Fisher’s LSD test.

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
