# Peer review of "High Light Acclimation Mechanisms Deficient in a PsbS-Knockout Arabidopsis Mutant"

_ijms, 2022, doi:10.3390/ijms23052695_

Round 1

Reviewer 1 Report

The authors advanced the information on the role of PsbS in acclimation processes under HL-stress conditions through specification of defects in long-term acclimation processes including upregulation of PQ biosynthesis genes in the PsbS-KO mutant of Arabidopsis. The results obtained is interesting, but the manuscript including Discussion section seems to have some inconsistent context and incorrect description, as describe below. Moreover, the role of PsbS in HL acclimation in the long-term is missing in the Discussion section. This reviewer recommends the authors to discuss the results consistently, with the role of PsbS under HL conditions proposed, and to revise the manuscript before acceptance in IJMS.

The reviewer’s concerns are as follows,

  1. 244

Data of PQ contents are not shown in this manuscript.

  1. 296

Was HL treatment period 3 h, or 2 h as described in the Fig. 6 legend?

  1. 307

Fig. 5A should be revised to Fig. 4A.

  1. 334

The phrase ‘Reduction in ROS’ is not proper. Fig. 6 shows that respective ROS increased under HL conditions even in the WT.

  1. 336-337, l. 79-81

The authors explain that ROS can be the signal to induce HL acclimation process in l. 336-337, but that inconsistently, ROS is the signal to block the induction of acclimation processes in l. 79-81. The authors should reconsider the manuscript context as to the effects of ROS.

  1. 339-340

The sentence ‘the failure of transcription induction of PQ biosynthesis genes may trigger some of acclimation processes’ seems to be incorrect. In the context, ‘trigger’ should be revised to ‘damage’ or some similar word.

Author Response

The reviewer’s concerns are as follows,

244

Data of PQ contents are not shown in this manuscript.

Response:

Thank you for the comment. As suggested, we removed “and PQ contents” [refer L256 in the revised ms].

296

Was HL treatment period 3 h, or 2 h as described in the Fig. 6 legend?

Response:

Thank you for the comment. As suggested, we revised “2h” [refer L310 in the revised ms].                   

307

Fig. 5A should be revised to Fig. 4A.

Response:

Thank you for the comment. As suggested, we revised to “Fig. 4A” [refer L321 in the revised ms].

334

The phrase ‘Reduction in ROS’ is not proper. Fig. 6 shows that respective ROS increased under HL conditions even in the WT.

Response: We have changed the phrase to:

“HL acclimation including the regulation in ROS homeostasis and the reduction in the light harvesting antenna size” [refer L347-348 in the revised ms]. We hope these changes are satisfactory.

336-337, l. 79-81

The authors explain that ROS can be the signal to induce HL acclimation process in l. 336-337, but that inconsistently, ROS is the signal to block the induction of acclimation processes in l. 79-81. The authors should reconsider the manuscript context as to the effects of ROS.

Response: We have changed the phrase to:

From these results, we suggest that PQ biosynthesis is one of the acclimation processes induced under HL stress, and the induction of many acclimation processes including PQ biosynthesis is blocked initially due to the over accumulation of ROS when the regulation mechanisms in ROS homeostasis are failed in npq4-1 mutants.” [refer L79-83 in the revised ms]. We hope these changes are satisfactory. We hope these changes are satisfactory.

339-340

The sentence ‘the failure of transcription induction of PQ biosynthesis genes may trigger some of acclimation processes’ seems to be incorrect. In the context, ‘trigger’ should be revised to ‘damage’ or some similar word.

Response: Thank you for the point.

We have revised the statement to “the failure of transcription induction of PQ biosynthesis genes may block the acclimation processes including the antenna size reduction”. [refer L354-355 in the revised ms]. We hope these changes are satisfactory.

Reviewer 2 Report

Presented work on arabidopsis PsbS-knockout mutant is highly significant for peer stress adaptation researchers to advance knowledge about the high light acclimation process, which remains to be poorly understood, and lesser-known. Please see the attached review report on the reviewed manuscript. 

Author Response

There are some minor issues reported in regard to typos, please see the review comments as given below.

Abstract: Abstract is written very succinctly and has rendered general significance and synopsis of the presented research, which could be useful for broader readership. However, authors have used six abbreviations which I think are way too many to be presented in abstract. Authors could avoid using abbreviations which are commonly used by peers such PSII, WT, and ROS.

Response:

We deleted three abbreviations that are not used more than two times in Abstract.

Introduction: Authors have provided well informed background about HL acclimation, PsbS protein and aptly relate it to the research problem. However, authors seem to have missed providing research hypothesis, which is important part of the research goal, and I would highly encourage authors to revise the introduction to consider the made suggestions.

Line 70-71: “we observed that the PsbS-deficient Arabidopsis plants failed to acclimate to long term HL illumination.”. This sentence is placed in the wrong section as this seems more like a discussion, please either rephrase it according to the desired intent of presenting introduction or move it in the discussion.

Response: Thank you for the point, and we deleted the sentence as shown in L70-71 in the revised ms.

Line 73-76: “GSEA evaluates microarray data at the level of gene sets. This gene set analysis is not useful for discovering new genes, but it is useful in discovering gene sets that have the same or related biological functions [24]. A coexpressed gene set is defined as a cluster of genes with similar expression patterns under various conditions [23]”. This sentence is placed in the wrong section as it is part of materials and method, please move it in the M&M.

Response:

We agree with the point raised by the reviewer, but we think it is also important to help the reader who is not familiar with GSEA and coexpressed gene sets etc., and therefore we would like to leave the sentences in a paragraph that appear early in Introduction.

Line 77-79: “From GSEA and qRT-PCR analyses, we observed that the expression of genes involved in PQ-9 biosynthesis was substantially reduced in npq4-1 mutants during long-term HL illumination resulting in a reduction in the PQ pool size.”. This sentence is placed in the wrong section as it is part of result, please move it in the results.

Line 79-82: “From these results, we suggest that ROS is one of the initial signals to block the induction of many

acclimation processes, and PQ biosynthesis is one of the acclimation processes induced under HL stress. However, npq4-1 mutants failed to reduce ROS accumulation and fail to maintain the PQ homeostasis” Must be moved to the discussion section. ......................................................................................................................

Response:

We agree with the point raised by the reviewer, but often we mention important parts of the results at the end of Introduction to help the reader to understand the purpose of the study and some important conclusions we draw in this study. Please allow us to keep this style and please refer L79-83 in the revised ms.

Results: Overall results are presented very eloquently and have explained in very detailed. Line 238: “genes” is repeated twice at the beginning of the sentence, please correct the same.

Response:

Thank you for the comment. As suggested, we corrected the error [refer L250 in the revised ms].                   

Discussion: Discussion is robustly discussed and highlighted the role of high light stress. Authors have also discussed their findings while comparing them with previously published research and identified potential interpretations those may help to future research.

Line 331: “increases” should be “increase”, please correct the same.

Response:

Thank you for the comment. As suggested, we corrected the error [refer L345 in the revised ms].                   

Materials and Methods: Authors have provided sufficient details on applied methodology that peer researchers can use to replicate the related experiments.

Conclusion: Few conclusions are provided in the discussion but still key take home messages are missing so authors are encouraged to write a small paragraph that provides key THM.

Response:

Thank you for the comment. So, we added a Conclusion section [refer L471-490 in the revised ms].                   

Reviewer 3 Report

Dear Authors,

Congratulations on submitting this nice work. I have read this manuscript with a good deal of zeal and zest. I found it worthy of publication. However i have some questions and suggestion.

The abstract should be revised. Eliminating the plentiful abbreviations and providing a quantitative overview of obtained results.

I found the introduction wordy. Highlight the novelty of this work, there should be a gap statement.

Authors should better develop a hypothesis.

Can the authors just describe the most significant results?

The titles and figure captions should be self-explanatory.

How the authors have selcted the time intervals for different parameters measurements? are not they redundanbt?

Can the authors refer to the most suitable studies on methods they followed for measurements?

Discussion is nicely drawn.

The manuscript lacks a concluding part. Please provide the conclusion section separately.

Author Response

The abstract should be revised. Eliminating the plentiful abbreviations and providing a quantitative overview of obtained results.

Response:

We deleted three abbreviations that are not used more than two times in Abstract.

I found the introduction wordy. Highlight the novelty of this work, there should be a gap statement.

Authors should better develop a hypothesis.

Can the authors just describe the most significant results?

Response:

We agree with the point raised by the reviewer and tried to modify the ms. However, we keep some parts such as the description of GSEA in the Introduction section to help the reader who is not familiar with GSEA and coexpressed gene sets etc. Please refer the modified ms and we hope these changes are satisfactory.

The titles and figure captions should be self-explanatory.

How the authors have selected the time intervals for different parameters measurements? are not they redundant?

Can the authors refer to the most suitable studies on methods they followed for measurements?

Response:

Thank you for the comment. As suggested, we modified figure captions and method sections as shown in the revised ms. We selected the time intervals that show the best difference between WT and mutants for the measurement of a specific parameter.

Discussion is nicely drawn.

The manuscript lacks a concluding part. Please provide the conclusion section separately.

Response:

Thank you for the comment. So, we added a Conclusion section [refer L471-490 in the revised ms].                   

Reviewer 4 Report

Paper "High light acclimation mechanisms deficient in a PsbS-knockout Arabidopsis mutant" is very interesting and important.

Authors investigated the role of PsbS protein during the HL acclimation processes in Arabidopsis. Authors compared several HL acclimation processes in WT and npq4-1 mutants, and they observed that the PsbS-deficient Arabidopsis plants failed to acclimate to longterm HL illumination.

Figure 1A needs regresion models and coefficients of determination.
Figures 2A and 2B need regresion models and coefficients of determination.
Figure 3A needs regresion models and coefficients of determination.
Figures 4A and 4B need regresion models and coefficients of determination.
Figure 5 needs regresion models and coefficients of determination.
Figure 6B needs regresion models and coefficients of determination.
Figures 7A and 7B need regresion models and coefficients of determination.
Figure 9 needs LSD of HSD values.

In the paper lack of "Statistical analysis" section. Paper needs major revision.

Author Response

Authors investigated the role of PsbS protein during the HL acclimation processes in Arabidopsis. Authors compared several HL acclimation processes in WT and npq4-1 mutants, and they observed that the PsbS-deficient Arabidopsis plants failed to acclimate to longterm HL illumination.

Figure 1A needs regresion models and coefficients of determination.

Response:

In Figure 1A, we added description in figure legends “Significance differences are indicated by asterisks (*P < 0.05; **P < 0.01; ***P < 0.001) compared to the WT as determined by simple linear regression analysis”.

Figures 2A and 2B need regresion models and coefficients of determination.

Response:

In figure 2 legend, we added description “Significance differences are indicated by asterisks (*P < 0.05; **P < 0.01; ***P < 0.001) compared to the HL 0h as determined by simple linear regression analysis”.

Figure 3A needs regresion models and coefficients of determination.

Response:

In figure 3A legend, we added description “Significance differences are indicated by asterisks (*P < 0.05; **P < 0.01; ***P < 0.001) compared to the WT as determined by simple linear regression analysis.”

Figures 4A and 4B need regresion models and coefficients of determination.

Response:

In figure 2 legend, we added description “Significance differences are indicated by asterisks (*P < 0.05; **P < 0.01; ***P < 0.001) compared to the HL 0h as determined by simple linear regression analysis”

Significance differences are indicated by asterisks (*P < 0.05; **P < 0.01; ***P < 0.001) compared to the WT (GL) as determined by simple linear regression analysis.

Figure 5 needs regresion models and coefficients of determination.

Response:

In Figure 5, we added description in figure legends “Significance differences are indicated by asterisks (*P < 0.05; **P < 0.01; ***P < 0.001) compared to the WT as determined by simple linear regression analysis”.

Figure 6B needs regresion models and coefficients of determination.

Response:

In figure 6B legend, we added description “Significance differences are indicated by asterisks (*P < 0.05; **P < 0.01; ***P < 0.001) compared to the WT (GL) as determined by simple linear regression analysis.”

Figures 7A and 7B need regresion models and coefficients of determination.

Response:

In Figure 7, we added description in figure legends “Significance differences are indicated by asterisks (*P < 0.05; **P < 0.01; ***P < 0.001) compared to the WT as determined by simple linear regression analysis”.

Figure 9 needs LSD of HSD values.

Response:

In Figure 9, we added description in figure legends “Significance differences are indicated by asterisks (*P 5 0.05; **P 5 0.01; ***P 5 0.001) compared to the HL 0h WT as determined by one-way ANOVA with Tukey test.”

In the paper lack of "Statistical analysis" section. Paper needs major revision.

Response:

Thank you for the comment. As suggested, we have performed regression analysis and added statistical analysis. We added a section in Materials and methods:

“4.11. Statistical analysis

Statistical analyses were performed using GraphPad Prism (v. 8.0). Significant differ-ences between experimental groups were analyzed by one-way ANOVA with Tukey’s HSD test or simple linear regression analysis, respectively. Detailed information about statistical analysis is described in the Figure legends. Statistical significance was set at P < 0.05. All experiments were repeated three to five times with similar results.”

Round 2

Reviewer 1 Report

The manuscript has been well devised.

Author Response

Thank you for the comment. English language of the revised 2nd round ms was thoroughly checked by a native English speaking scientist.

Reviewer 4 Report

Authors added: “Significance differences are indicated by asterisks (*P < 0.05; **P < 0.01; ***P < 0.001) compared to the WT (GL) as determined by simple linear regression analysis.” or “Significance differences are indicated by asterisks (*P < 0.05; **P < 0.01; ***P < 0.001) compared to the HL 0h WT as determined by one-way ANOVA with Tukey test.” but in all Figures still lack of "regresion models and coefficients of determination" as well as "LSD or HSD values".

"Statistical aalysis" is very poor. Lack information about districution of observed traits. 

Paper needs major revision.

Author Response

Thank you for the comment. We tried our best to do statistical analysis as it is suggested.

English language of the revised 2nd round ms was thoroughly checked by a native English speaking scientist.

Round 3

Reviewer 4 Report

Now, all is ok.